# DMET^TM^ Genotyping: Tools for Biomarkers Discovery in the Era of Precision Medicine

**DOI:** 10.3390/ht9020008

**Published:** 2020-03-29

**Authors:** Giuseppe Agapito, Marzia Settino, Francesca Scionti, Emanuela Altomare, Pietro Hiram Guzzi, Pierfrancesco Tassone, Pierosandro Tagliaferri, Mario Cannataro, Mariamena Arbitrio, Maria Teresa Di Martino

**Affiliations:** 1Department of Medical and Surgical Sciences, Magna Graecia University, 88100 Catanzaro, Italy; agapito@unicz.it (G.A.); marzia.settino@studenti.unicz.it (M.S.); hguzzi@unicz.it (P.H.G.); cannataro@unicz.it (M.C.); 2Department of Experimental and Clinical Medicine, Magna Graecia University, Salvatore Venuta University Campus, 88100 Catanzaro, Italy; scionti@unicz.it (F.S.); emanuela.altomare@gmail.com (E.A.); tassone@unicz.it (P.T.); tagliaferri@unicz.it (P.T.); 3CNR-Institute for Biomedical Research and Innovation, 88100 Catanzaro, Italy

**Keywords:** pharmacogenomics, SNPs, DMET^TM^ platform, GMQL, bioinformatic tools

## Abstract

The knowledge of genetic variants in genes involved in drug metabolism may be translated into reduction of adverse drug reactions, increase of efficacy, healthcare outcomes improvement and economic benefits. Many high-throughput tools are available for the genotyping of Single Nucleotide Polymorphisms (SNPs) known to be related to drugs and xenobiotics metabolism. DMET^TM^ platform represents an example of SNPs panel to discover biomarkers correlated to efficacy or toxicity in common and rare diseases. The difficulty in analyzing the mole of information generated by DMET^TM^ platform led to the development and implementation of algorithms and tools for statistical and data mining analysis. These softwares allow efficient handling of the omics data to validate the explorative SNPs identified by DMET assay and to correlate them with drug efficacy, toxicity and/or cancer susceptibility. In this review we present a suite of bioinformatic frameworks for the preprocessing and analysis of DMET-SNPs data. In particular, we introduce a workflow that uses the GenoMetric Query Language, a high-level query language specifically designed for genomics, able to query public datasets (such as ENCODE, TCGA, GENCODE annotation dataset, etc.) as well as to combine them with private datasets (e.g., output from Affymetrix® DMET^TM^ Platform).

## 1. Introduction

In the era of precision medicine, the identification of germline variants related to the inter-individual variability observed in response to the same drug represents a great opportunity for evidence-based drug prescription. Individual drug response is influenced by multiple and highly variable factors which impact on the pharmacokinetic (PK) and/or pharmacodynamic (PD). In fact, both physiological and pathological conditions (aging, kidney and liver function, comorbidities, environmental conditions) together with genetic background may interfere with drug efficacy and/or toxicity. Although many studies have identified germline polymorphic variants in genes involved in Adsorption, Distribution, Metabolism and Excretion (ADME) of drugs in relation to inter-individual variability, their identification as part of common clinical practice remains an active challenge. In the post genomic era the terms pharmacogenetics and pharmacogenomics are interchangeable and are here referred to as PGx. The objectives of PGx include safer prescriptions, powerful and appropriate drugs, reduction of healthcare costs and treatment toxicities. In the field of oncology, the role of biomarkers for the development of precision medicine provides a strategic opportunity for technological developments to improve human health and reduce health-care costs. While oncogenic somatic alterations are well known to influence drug response, germline mutations, such as those in *BRCA*, *PALB2*, *ATM*, and *CHEK2* genes, were thought to simply determine individual predisposition to develop breast, ovarian and other cancers [1,2,3]. However, among germline alterations, the polymorphic variants in genes encoding for drug-metabolizing enzymes can be used as predictive biomarkers for drug efficacy and drug-induced toxicity [4]. Therefore, germline and somatic variants are both potential PGx predictive biomarkers for drug targets, drug related toxicity and inter-individual variability in drug response, in the perspective of precision medicine. In PGx studies, the genetic variants evaluated are Single Nucleotide Polymorphisms (SNPs), nucleotide insertions, deletions, short tandem repeats, copy number variations (CNV) and chromosomal translocations [5]. SNPs are common inherited variations (90%) that exert a biological role only when they occur within a regulatory region or in a gene coding sequence. SNPs are inherited within haplotype blocks and exist in strong linkage disequilibrium (LD) with a specific genetic variant. In other words, the identification of a specific SNP allows one to assume the presence of the co-inherited genetic variant. For this reason, SNPs can be used as tag markers for unseen causative alleles. The closely linked SNP alleles are in blocks, separated by regions of high recombination (hotspots), in which there are few or many polymorphic variants which also could be associated with a disease or drug-response phenotype [6]. Keeping this in mind, the genotype-phenotype correlation assumes high importance in PGx studies and in determining the individual response to pharmacological therapy.

The identification of polymorphic variants with impact on phenotype has evolved through three approaches: (1) the candidate gene approach, where a small number of well-known PK or PD-related markers are tested in a small sample size; (2) the genome wide association study (GWAS) approach, a hypothesis-free method performed on large populations where high numbers of markers are simultaneously tested, identifying only common variants but with the need of stringent statistical correction [7]; (3) the pre-defined SNPs panel approach which includes only thousands of candidate SNPs relevant pharmacogenes with putative importance. Advancement in technologies, such as next generation sequencing (NGS), contemplate a diagnostic tool where whole genome or exome sequences are interrogated as comprehensive PGx genotyping tool in a rapid and large-scale DNA sequencing technology [8]. Therefore, there has been a revolution in tumor treatment prescription and in drug’s labeling PGx recommendation towards a more tailored therapy. The candidate gene approach has the limitation of high rate of false positives, replication of results and overestimation of effect size, while the multiple comparisons correction used in GWAS leads to false negatives. In addition, GWAS arrays have a good coverage of PK genes but include a limited number of known PD genes. The SNP panel represents an ideal compromise between the other two approaches through the simultaneous genotyping of SNPs with known relevance in PK-PD and a limited need for statistical stringency for multiple comparisons [9]. Users have the option to customize a SNP-panel choosing the candidate genes and SNPs that suite best their needs.

The DMET^TM^ platform represents an example of the SNPs panel widely used to discover biomarkers correlated to efficacy or toxicity in common and rare diseases [10,11,12,13,14,15]. The difficulty in analyzing the mole of information generated by DMET^TM^ platform led to the development and implementation of algorithms for statistical as well as data mining analysis. Moreover, in order to integrate and validate big-data derived from omics technologies, high-performance software tools, that are able to deal with these vast amounts of data, are necessary. Software tools allow efficient handling of omics data to validate the explorative biomarkers identified by DMET assay and to correlate them with drug efficacy, toxicity and/or cancer susceptibility. These analyses should lead to identifying potentially predictive biomarkers to be confirmed in well-designed clinical trials for translating purposes. In this review, we aimed to describe software tools developed specifically for DMET genotyping analysis. In addition, we describe how to employ the available software tools in order to integrate information derived from analysis of matched data (disease, treatment, stadiation, race). The latter might represent a new approach for biomarkers validation.

## 2. DMET^TM^ Genotyping Platform 

The DMET™ Plus array (Thermo Fisher Scientific, Inc., Waltham, MA, USA) is a PGx high-throughput genotyping assay, based on microarray technology. It allows to analyze comprehensively, by a single multiplexed assay, 1931 SNPs and five copy number variations (CNVs) located in 231 genes encoding phase I and II metabolism enzymes, drug transporters, drug targets and modifiers, approved by Food and Drug Administration (FDA) as involved in drugs and carcinogenic metabolism (Appendix A) [16]. The DMET^TM^ platform is recognized for research application only. The assay starts from 1 μg of genomic DNA extracted from peripheral blood, formalin fixed-paraffin-embedded tissue, or buccal swab. The DMET^TM^ Plus assay uses Molecular Inversion Probe (MIP) technology [17] to amplify the sequence-specific targets for each 1936 marker. MIPs are single-stranded oligonucleotides consisting of a common linker containing universal PCR primer binding sites flanked by target-specific probes. The assay probe pool contains one or more MIPs for each of the genotyping markers. The target-specific probes simultaneously hybridize to the same DNA fragment, forming a circular structure with the intended target captured between the probes (annealing); addition of polymerase and ligase results in gap filling and completion of the circular form by incorporation of the intervening target sequence (gap fill and ligation). Library formation is performed by cleaving the circular form (exonuclease digestion and inversion), followed by standard multiplex PCR. PCR products were than fragmented, labeled with biotin and hybridized overnight (16–18 h) to the array. Finally, arrays were scanned to generate the file containing the intensity signal for all probes. 

After normalization of raw signal values, genotypes are generated using the Affymetrix® DMET Console software (Version 1.3, Thermo Fisher Scientific, Inc., Waltham, MA, USA) as single-sample genotyping and converted to standardized name (star nomenclature) to track known polymorphic variants. Genotypes are determined for each SNP site and reported as homozygous wild-type, heterozygous, homozygous variant, ‘no call’ or Possible Rare Allele (PRA) in the case of lack of genotype call. The reproducibility of genotyping results with Sanger sequencing or TaqMan SNP genotyping assays is 100% [18]. The DMET^TM^ Console generates CEL files containing the raw data produced by chip-array scans. The CEL files are translated in CHP files and after are exported as xlsx (excel) or plain text files, formats suitable for statistical or data mining analysis. The analysis of microarray data is conducted through four steps: (a) preprocessing, which allows background correction, summarization and normalization; (b) annotation (process which associates to each gene a set of functional information) and translation (process which converts the genotype calls (reported in CHP files) to tracked functional allele calls); (c) statistical/data mining analysis; and (d) biological interpretation. The conversion of intensity value in actionable knowledge can be conducted by DMET^TM^ Console, apt-DMET-genotype and DMET Analyzer. Affymetrix® DMET Console and apt-DMET-genotype allow the summarization/normalization of CEL files, the annotation and translation of genotype calls, then the CHP file and ARR sample files are merged and annotated using standardized nomenclature. DMET-Analyzer allows the automatic analysis of data provided by Affymetrix® DMET Console [19]. A scheme of the DMET analysis workflow is represented in Figure 1.

## 3. Software Tools to Analyze Genotyping Data

Software tools have become needful to support researchers in omics science and to efficiently process the vast amounts of data produced by high-throughput technologies. The continuing improvements in microarrays and NGS mainly due to the continuous reduction of costs, along with the improvement in the number of variants and/or genes that can be investigated in a single experiment, spur for the development of software frameworks able to transform the available amount data in actionable knowledge. The development of appropriate software frameworks and algorithms for the efficient and scalable data analysis will allow researchers to analyze considerable amounts of genotyping data in a short time as well as to provide enhancement in the accuracy of the results due to the capability to investigate the problems from a boarder perspective. Thus the most suitable software tools can contribute to move the first steps towards the so called Predictive, Preventive, Personalized and Participatory Medicine (P4 medicine) [20].

Although vendors of genomics platforms provide its proprietary software frameworks with which it is possible only to handle raw data, i.e., convert the intensity data file in gene expressions or SNPs value. Vendors do not offer any software tools capable to extract actionable knowledge from the data.

Algorithms and software frameworks able to manage the converted, annotated and summarized genotyping raw data, they can support researcher in the phase of data analysis to identify knowledge buried into the data and not easily available. For example, in a case-control study, to identify the SNPs related to an adverse drug reaction into the population under investigation, without the support of a specific analysis framework, the analysis should be done manually, a long and error prone task considering the huge amount of data that must be analyzed. Instead, the use of specific software frameworks can speed up and simplify the knowledge extraction, since can be accomplished automatically.

In this section we present suite of bioinformatic frameworks for the preprocessing and analysis of DMET-SNPs data. The Affymetrix® DMET^TM^ platform allows us to identify genes involved in ADME functions obtained in case-control studies. Below we review the open-source bioinformatics frameworks developed at the Bioinformatics Laboratory of the University of Catanzaro, along with five other tools providing an overview of their main technical and functional features [21]. A description of the investigated software tools is presented below. Furthermore, a methodology to integrate genomic data by using GMQL-Web and a case study are provided.

*DMET-Analyzer* [22] is a software framework for the automatic association analysis between the variations present in the patients’ genome and their clinical conditions, i.e., the different response to drugs. DMET-Analyzer simplifies the identification of relevant variations into the population under investigation through the automation of the whole statistical analysis workflow. Moreover, DMET-Analyzer can annotate SNP data by automatically retrieving the information available in existing SNP databases, e.g., dbSNP, as well as to interpreter the biological process in which variants are involved through the association of SNP within a pathway, by automatically retrieving the information stored in specialized databases, e.g., PharmaGKB. DMET-Analyzer is written in Java making it compatible with Unix/Linux, MacOS and Windows operating systems, it presents a simple graphical user interface that allows non-programming users (e.g., doctors and biologists) to analyze DMET files interactively produced using the Affymetrix DMET-Console. Moreover, DMET-Analyzer implements the FDR and Bonferroni statistical correctors, the Odds-Ratio and HardyWeinberg equilibrium calculator. DMET-Analyzer can analyze only DMET files in xlsx (i.e., excels files) or plain tab delimited format (i.e., plain text files). Results can be easily saved by clicking on it and saved in textual format (txt), markup language (html) and so on. DMET-Analyzer is freely available under the GNU General Public License version 2.0 [23].*DMET-Miner* [24] is a software tool for mining association rules from DMET SNP datasets. DMET-Miner through the association rules can correlate the presence of multiple allelic variants with the clinical condition of the patients. Allowing the users to overcome the limitation of the univariate statistics implemented in DMET-Analyzer that can extract associations among a single allelic variant and the clinical conditions of samples. For example, the most frequent association among alleles responsible for the different response to a treatment. DMET-Miner enables users to automatically mine association rules from a whole DMET datasets, conversely from other available tool e.g., Weka requiring to the user to preprocess the input file in order to handle missing value, trivial data and so on. DMET-Miner is written in Java, making it available for all the operating systems compatible with Java. It presents a simple graphical user interface, allowing the users to analyze a dataset through some mouse’s clicks. DMET-Miner is distributed under Creative Commons license and is freely downloadable for academic and not-for-profit institutions [25].*OS-Analyzer* [26] is a software framework implemented in Java for the analysis of SNP microarray datasets enriched with survival events. OS-Analyzer comes with a simple, effective and intuitive graphical user interface for the automatic computation and visualization of Overall Survival (OS) and Progression Free Survival (PFS) curves of case patients, evaluating their association with ADME gene variants. Moreover, to simplify the researchers work, results according to statistical significance obtained by comparing the area under the ROC (Receiver Characteristic Curve) curves are ranked. Statistical Relevance is computed by using the log-rank test, allowing a quick and easy analysis and visualization of high-throughput data. OS-Analyzer is distributed under Creative Commons License, is freely available for academic and not-for-profit institutions [27].*Affymetrix® DMET Console* allows to preprocess the raw data file generated by the Affymetrix DMET for building a comprehensive table containing, for each probe and for each sample, the detected SNP or a No call value (i.e., ambiguous nucleotide in the SNP). DMET Console support probe-set summarization of a complete dataset of binary “.CEL” files (containing the probe-level intensities), the management of resulting preprocessed files “.CHP” (containing the gene-level information) and the building of tabular dataset containing the genotype call for all the probesets and all the samples of an experiment. Once the preprocessing phase is completed, the relationship between the detected SNPs and the response to drugs must be tested. Because of DMET Console does not allow this test, in order to discover statistically significant associations, researches have to export and manually process SNPs tables produced by the DMET Console thought the use of external tools (e.g., statistical software).*Affymetrix Power tools suite* (APT) is a set of command line programs that implement different algorithms for preprocessing Affymetrix microarray data. Two of the most popular programs in APT are: apt-dmet-genotype, for making genotype calls from Affymetrix genotyping arrays, and apt-probeset-summarize for analyzing gene expression arrays. Such as DMET Console, both programs are generally focused on .CEL file analysis.*The GenoMetric Query Language* (GMQL) [28] is a high-level query language, inspired by classic traditions of data-base management, that extends conventional algebraic operations with bioinformatics domain-specific operations specifically designed for genomics; thus, it supports knowledge discovery across thousands or even millions of samples comparing genomic regions on the basis of metric properties but also arbitrary attributes and metadata that concern regions and samples, respectively. In particular, datasets are described by the Genomic Data Model (GDM) [29], based on the notion of genomic region, which provides interoperability between several data formats. In addition, GDM combines abstractions for genomic region data with the associated experimental, biological and clinical metadata. GMQL system can be used online through a specific Web interface which provides a user-friendly intuitive environment for bioinformaticians and biologists who need to query genomic processed data (e.g., sourcing from big consortia such as ENCODE, TCGA, GENCODE or RefSeq) as well as to combine them with their private datasets (i.e., datasets created by a specific user as result of their own experiments and studies, e.g., output from Affymetrix® DMET^TM^ Platform). Such environment provides portable and scalable genomic data management on powerful servers and clusters (based on Apache Spark).*PLINK* [30] is a free, open-source tool for GWAS and research in population genetics. PLINK works on five core functional domains: data management, summary statistics, population stratification, association analysis and identity-by-descent estimation. Association tests can be run by PLINK to evaluate case-control data to determine if an SNP has an effect on disease status. PLINK can run either as a stand-alone tool (from the command line or via shell scripting) or in conjunction with gPLINK, a Java-based graphical user interface (GUI) that offers a simple project management framework to track PLINK analyses and facilitates the integration with Haploview.*Haploview* [31] is a comprehensive suite of tools, written in Java language, for analysis and visualization of LD and haplotype maps. Haploview accepts input in a variety of formats and generates marker quality statistics, LD information, haplotype blocks, population haplotype frequencies and single marker association statistics. Haploview is fully compatible with data dumps from the HapMap project. HapMap genome browser allows researchers to explore a particular region of the genome and extract HapMap genotype data for all genotyped markers in the selected region in a format accepted by Haploview. Haploview currently supports visualization and plotting of PLINK whole genome association results [32].

The significant improvements because of the use of the microarrays have led to an increase in the number of variants that it is possible to investigate in a single experiment as well as a reduction of the analysis time, allowing to produce considerable amounts of data in ever shorter times. Therefore, it is necessary to develop software tools and algorithms capable of managing ever-increasing volumes of data. The following is a summarization of the parallel and web applications developed to efficiently deal with huge genotyping datasets.

*coreSNP* [33] is a tool implemented in Java language, for the parallel pre-processing and statistical analysis of DMET SNP datasets. The scalable implementation is obtained exploiting the multi-threading capabilities of modern CPUs, allowing core SNP to manage huge amount of DMET data. The automatic association analysis between possible genome variations of the patient and the clinical conditions through the well-known Fisher’s Test is obtained. Moreover, multiple-statistical correctors i.e., Bonferroni and False Discovery Rate, with which to improve the statistical significance of results, are available. The visualization of the detected SNPs as heatmap plot provides a visual feedback that simplifies the interpretation of the results.Parallel Association Rules Extractor from SNPs (*PARES*) [34] is a multi-thread software tool developed in Java for the parallel extraction of association rules by which to correlate the presence of a multiple allelic variants with the patients’ clinical condition, i.e., the most likely set of alleles responsible for the onset of adverse drug reactions. PARES is a multi-thread version based on the optimized version of the Frequent Pattern Growth (FP-Growth) algorithm. PARES encompasses a customized SNP dataset preprocessing approach based on a Fisher’s Test Filter to prune trivial transactions, allowing to shrink the search space as well as to reduce the FP-Tree size enabling a better management of the main memory. PARES comes with a simple and intuitive graphic user interface, where specific skills are not necessary to extract multiple relations between genomic factors buried into the datasets. PARES is distributed by BioinfoLabUnicz, under Creative Commons license, is freely downloadable for academic and not-for-profit institutions [35].*GenotypeAnalytics* [36] is a representational state transfer (RESTFul) service by which to mine association rules from SNP-dataset through the use of a common web-browser. GenotypeAnalytics can speed up and simplify the analysis of massive amount of SNPs data, highlighting in a remarkable way only the SNP involved in the development of the disease or in adverse drug reaction. GenotypeAnalytics can support researcher and medical doctors to discern new molecular markers that can be (after further opportune validations) used into the clinical contexts. Possible new molecular markers can be used to perform pathway enrichment to understand which SNPs are responsible for the pathway’s anomalies or the SNPs that influence drug responses in subjects with the same pathology following the same therapy.*Cloud4SNP* [37] is a Cloud software tool for the parallel pre-processing and statistical analysis of DMET SNPs data. Cloud4SNP is the Cloud version of DMET-Analyzer [22] that has been implemented on the Cloud using the Data Mining Cloud Framework, a software environment for the design and the execution of knowledge discovery workflows on the Cloud. Cloud4SNP is developed in Java and presents a simple graphical user interface that can be accessed by means of a common web-browser. Providing the analysis workflow as a service, it allows the users to upload and analyze the data without to buy expensive hardware or to setup the analysis environment. Cloud4SNP allows one to identify the relevance of the presence of SNPs in one of the two classes of samples using the well-known Fisher test, along with the use of multiple-statistical correctors such as Bonferroni and FDR.

The main features of the presented software frameworks are summarized in Table 1.

## 4. A New Methodology to Integrate Genomic Public Data with Genotyping Private Data

The developments of high-throughput technologies have accelerated the accumulation of massive amounts of omics data from multiple sources. In contrast to the traditional approaches in which data from each source have been analyzed in isolation, integrative analysis of multi-omics and clinical data has become a key for new biomedical discoveries and advancements in precision medicine [20]. Indeed, isolated omics studies frequently fall short when identifying the cause of complex diseases such as cancer. Although there has been much technological progress, the integration across data sets and data types remains limited. In this section we illustrate a methodology [38], based on the use of GMQL System, for combining private datasets such as DMET Platform [19] output with public datasets already available in the GMQL Repository as well as external (i.e., external public dataset can be uploaded following the steps of workflow in Figure 1 such as for private dataset). A private dataset is an alternative to public dataset and it is created by a specific user through some uploading operations or it is a result of some GMQL queries. The methodology consists of the following main three phases: data preprocessing, data ingestion and query-download results (Figure 2).

### 4.1. Data Preprocessing Stage

This stage prepares raw data for further processing and it is necessary since most of the datasets obtained as outputs from various systems are noisy, incomplete, inconsistent and contain outliers. Data preprocessing is a method of resolving such issues. Once preprocessed, the dataset needs to be transformed into a GDM compliant format in order to employ it within GMQL-web. Each GDM dataset is a collection of samples with the same region data schema and with the same type features; each sample, in turn, corresponds to a pair of files which contain:

(i) region data: describing the physical coordinates of the genomic areas (e.g., the genomic position of the Affymetrix probes) and other optional genomic features (e.g., p-value, peak, q-value);

(ii) metadata: unstructured attribute-value pairs describing general properties (i.e., biological, clinical, and experimental) of the sample;

It must be considered that conversion of genome coordinates between different assemblies is often required for many integrative and comparative studies (e.g., Affymetrix DMET results are aligned to GRCh37/hg19 genome build thus they could need to be remapped to current genome build GRCh38.

### 4.2. Data Ingestion Stage

GMQL-Web interface allows one to combine private datasets with public datasets through the “Add/Upload new dataset” feature that allows the user to choose among two options for uploading private datasets files: the *standard file-format mode* allows one to use a number of file standard formats directly supported by the system [39] (e.g., BED, NarrowPeak, BroadPeak, VCF, etc);the *custom file-format mode* can be chosen by selecting the “Custom (GTF or tab/delimited)” option and it allows one to use a user-defined format following the guidelines of Gene Transfer Format (GTF) [40] or TAB-delimited formats. In this case it is required the definition of an additional Extended Markup Language (XML) format Schema file describing the structure of the dataset to upload. Once uploaded, the private datasets are shown in the “interface datasets viewer” under the “Private” folder and it can be managed independently or in combination with other public datasets.

### 4.3. Query/Download Results Stage

A GMQL query is a sequence of operations, applied on one or more datasets, resulting in the creation of new datasets. A query is expressed as a sequence of GMQL operations, each with the following structure:

<OV> = OPERATOR (<params>) <IV1><IV2>

Where OV stands for output variable, an operator can be tuned using optional parameters, and IV1 and IV2 are input variables (Data-Driven Genomic Computing [41]).

Among all GMQL unary and binary operators, we here focus only on three of them, since they are used in the reported following example:

SELECT creates a new dataset from an existing one by extracting a subset of samples from the input dataset. Several conditions can be combined using Boolean operators;

MATERIALIZE writes the content of a dataset to a file and it registers the saved dataset in the repository to make it usable in other queries;

MAP applies to two datasets, a reference and an experiment. For each sample in the experiment dataset, it computes aggregates over the values of the regions that intersect with at least one region, in at least one reference sample. The count aggregate counts the number of experiment regions that intersect a specific reference region.

The right side of Figure 1 shows GMQL queries can be applied to public datasets optionally combined with private ones (derived from the previous preprocessing and ingestion phases). The GMQL Web “Query editor” allows one to write the query that will be later compiled and executed. Once results are produced (i.e., materialized) they become available in the “Datasets viewer” for browsing, further processing or downloading.

## 5. Case Study

### 5.1. Integration of Private DMET Platform Data with GENCODE Annotation Database

In this Section we illustrate a representative example of how to applicate our methodology for integrating DMET platform genotype results, with dataset sourcing from GENCODE annotation database [42]. The methodology used the workflow shown in Figure 3 involving DMET platform, GMQL system and, in general, public genomic databases. Figure 4 focuses on the case study workflow that specifically involved GENCODE as a public genomic database.

### 5.2. Data Preprocessing Stage

DMET-Console output is a tab-delimited file structured as a matrix with 1936 rows (probes) and a number of columns related to the number of subjects enrolled in the analysis. The value contained in the (*i-th,j-th*) cell is the SNP detected in the *i-th* probe and belonging to the *j-th* subject. The tab-delimited file obtained from DMET Console needs be transformed into a GDM compliant format. We located the position of the DMET SNPs within the genomic region by using the DMET [43] annotation file which provided the genomic coordinates for each Affymetrix probe. In order to integrate the DMET SNPs output with the GENCODE v27 dataset, already available in the GMQL Repository, the DMET probes genomic coordinates were mapped to current genome build GRCh38 by using the UCSC Lift Genome Annotations tool [44].

### 5.3. Data Ingestion

As depicted in Figure 2 (central part) the user can choose among two options for uploading private datasets files. We have chosen the *custom file-format mode*. We created a tab-delimited file containing the DMET dataset where the columns are defined as follows: the first is the Affymetrix Probe identifier (i.e., a unique Affymetrix identifier for the probe set or SNP); the next four contain the values describing the genomic regions (i.e., chromosome, start, stop, strand); the fifth and sixth columns contain respectively the dbSNP RS ID (i.e., RS ID from dbSNP database, e.g., rs11584174, rs2501870, etc.) and the associated identifiers in the PharmGKB database. The remaining columns contain the detected SNPs for each subject. The XML schema file describing the structure of the dataset must be created before loading the DMET dataset. Once the dataset file and its schema are obtained, we uploaded them in GMQL-Web using the feature “Add/Upload”.

### 5.4. Query/Download Results

Once the DMET output file (containing the SNPs detected by 33 probes on 28 subjects) was uploaded into GMQL Web system, the query in Listing 1 (Figure 5) was executed in order to:select SNPs from DMET Dataset;select exon regions from GENCODE annotation dataset;map exon regions on SNPs regions;select only SNPs regions that overlap at least one exon region.

The result of the MAP operation (EXON_Map) was a dataset which contained SNPs equipped with the count (SNP_count) of exons overlapping with that nucleotide. Many SNPs did not overlap with exon, thus the SNP_count is zero. The select statement filters out samples containing a region with null count. Results of this query (Figure 6) contain SNPs region coordinates (chromosome, start and end positions, strand) and additional characterizing attributes (gene name, dbSNP ID, all SNPs detected by DMET in that region and the number of overlapped exon regions).

## 6. Conclusions

Comprehension of the role of common variations in ADME genes has the potential to significantly improve clinical research by predicting the contribution of an individual genetic make up to PK and PD activity. This knowledge may be translated into reduction of ADRs, increase of efficacy as well as healthcare outcomes improvement and economic benefits. Many high-throughput tools are available on the market for the genotyping, at different scale, of SNPs known to be related to drugs and xenobiotics metabolism. Among these the DMET technology has been widely used in clinical research. Understanding how genes are affected by SNPs associated with a pharmacological phenotype is crucial for the prescription of the right drug for the right disease in the era of precision medicine. The advancement in high-throughput technologies has urged to consult and integrate massive amounts of public omics data from different sources as well as to validate private genotyping results. For instance, DMET plus SNPs platform applied to a limited and private sample of patients, allows one to identify genotypes that may support the discovery of biomarkers in the optic of precision medicine. 

However, genetic associations identified in the initial study need to be confirmed in an additional non overlapping study samples in order to replicate the statistically correlation between the same genetic variants and the trait of interest. Moreover, the biomarker validation process includes two important concepts i) analytical validation and ii) clinical qualification. The first refers to the reanalysis of all, or a subset of genetic variants investigated in the initial study, and includes the process to assess the accuracy, the robustness and reproducibility of the assay. The clinical qualification regards the correlation of the identified biomarker with biologic processes and clinical endpoints. The clinical translation of a biomarker is regulated in the European Community (CE) and United States (US) by different steps to achieve the inclusion in clinical practice guidelines [45] as FDA-cleared or CE-IVD marked clinical diagnostic tests. In this scenario, annotated SNP effects need to be interpreted in the broader context of genes, LD and molecular pathways, and so functional information could be integrated from different repositories. The integration of genotype data with biological and clinical information, could be possible using public databases including gene expression, methylation and sequencing data. The functional validation of the impact of a genetic variant, hypothesized through the integration of GWAS/exome, RNA-seq, and protein data, will contribute to reveal further translational and protein structural effects of potential biomarkers. 

In this work we described all the algorithms developed up to now to analyze genotyping data, generated through DMET Console software and GMQL System, with the aim to integrate private DMET Platform results, across data sets and data types output, with public datasets already available in the GMQL Repository and external public dataset. In our model of analysis, the opportunity to insert specific queries to interrogate a public dataset with the aim to identify predictive biomarkers of drug efficacy/toxicity or disease susceptibility, allows us to characterize also the role of tag SNPs providing useful confirmation of hypothesized models for gene and genome dynamics. The integration of genotype data with biological and clinical information through the approaches described is important for the understanding, expansion and validation of the knowledge on the SNP-association and its clinical translation. The identification of predictive biomarkers is the major goal of biomedical research. However, the ability to consult and integrate large-scale molecular data have not been largely applied, precluding further advances in precision medicine. Additional efforts are needed to translate into clinical practice the promise of genomic and other molecular data.

## Figures and Tables

**Figure 1 high-throughput-09-00008-f001:**
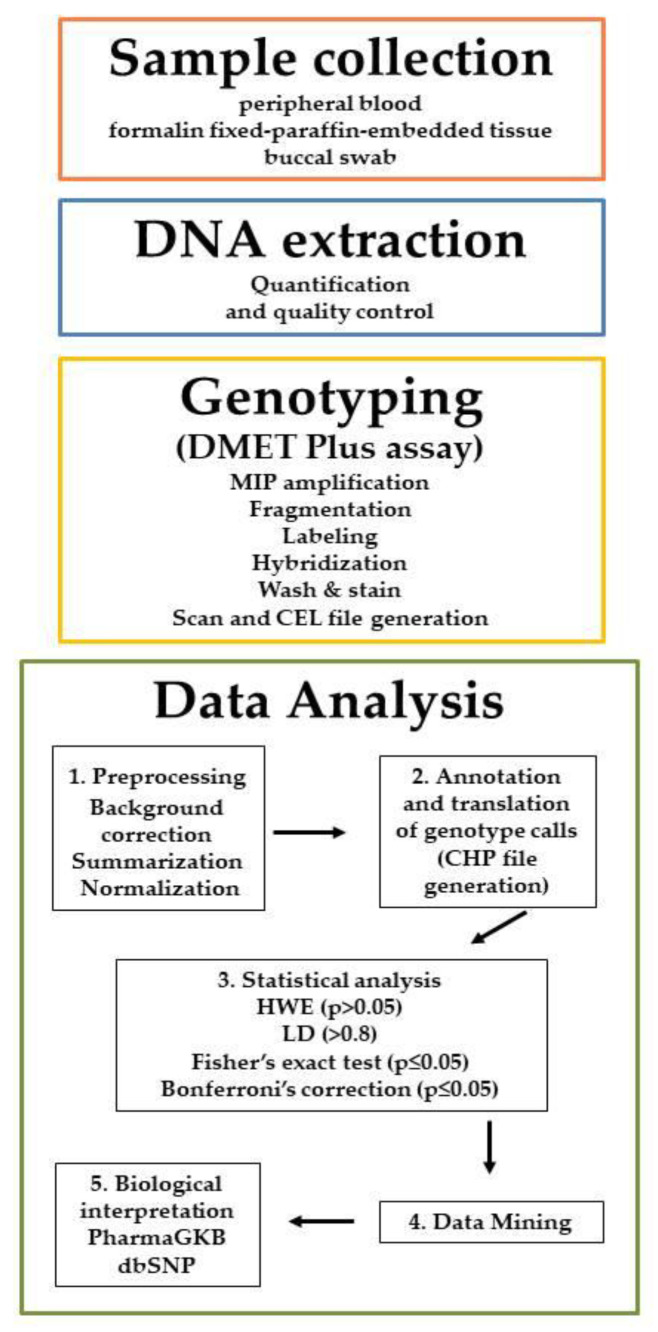
DMET analysis workflow.

**Figure 2 high-throughput-09-00008-f002:**
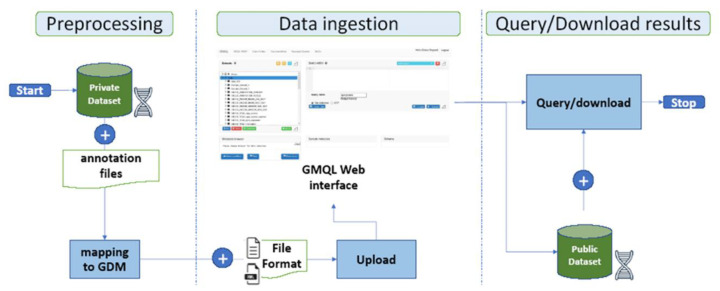
Phases of the methodology for integrating private and public datasets using GMQL-Web interface data preprocessing stage.

**Figure 3 high-throughput-09-00008-f003:**
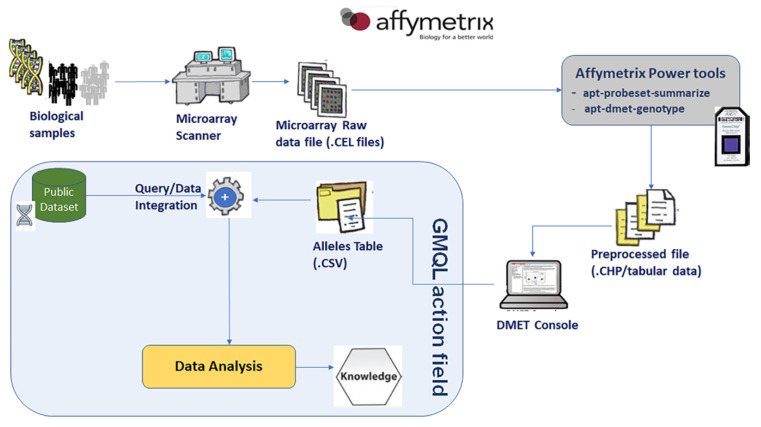
Workflow involving DMET platform, GMQL system and public genomic database (e.g., GENCODE).

**Figure 4 high-throughput-09-00008-f004:**
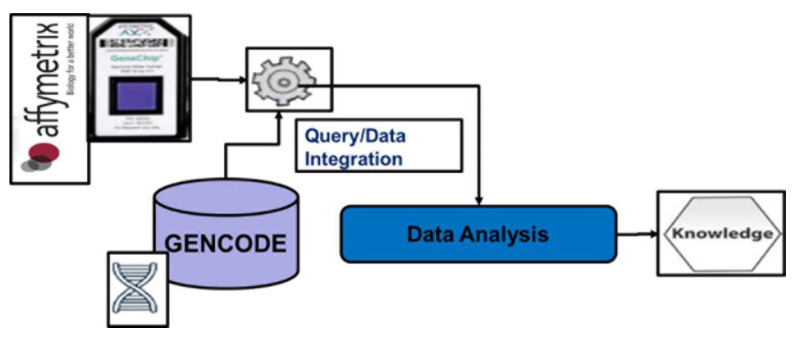
Example of application of the methodology: integration of Affymetrix DMET platform outcome with GENCODE annotation database.

**Figure 5 high-throughput-09-00008-f005:**
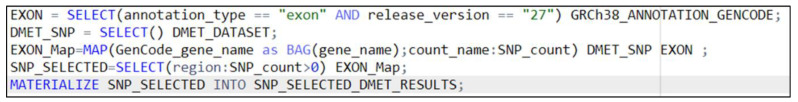
Listing 1—query for integrating DMET dataset with GENCODE dataset.

**Figure 6 high-throughput-09-00008-f006:**
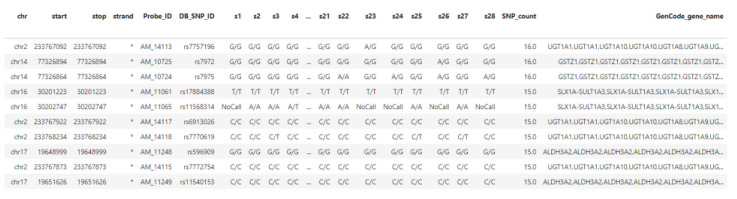
Results of Listing 1.

**Table 1 high-throughput-09-00008-t001:** Main features of the presented software frameworks.

Software	Availability	Main Features	IA	Interface	OS	DM	Statistical	Prep	CS	Weaknesses	Strengths
DMET-Analyzer	Free	Analysis of Variants based on Statistical test	Annotations and direct link to external databases	Command line, GUI	Platform-independent	NO	YES	YES	L	Single allelic variants discovery; Unable to dig with Affymetrix raw files;	It automatically analyses data in case-control association studies;
DMET-Miner	Free	Analysis of Variants using Association Rules mining	Annotations and direct link to external databases	GUI	Platform-independent	YES	NO	YES	L	Manage small Dataset	Multiple allelic variants discovery
OS-Analyzer	Free	computation and visualization of OS and PFS curves	Integration of genomic and clinical data	GUI	Platform-independent	NO	YES	YES	L	It cannot analyse gene expression data in order to plot OS curves (3)	High performance with respect to other statistical tools; Automatic analysis of whole DMET SNPs Dataset
Affymetrix DMET-Console	Free*	Preprocessing of raw data generated by DMET	NO	GUI	Windows	NO	NO	YES	L	Lacks in the possibility of doing statistical and data mining analysis;	It allows only the preprocessing of binary data
Affymetrix Power tools suite (APT)	Free*	They implement algorithms for analyzing and working with Affymetrix Microarrays	YES	command line	Platform-independent	NO	NO	YES	M	Lacks in the possibility of doing statistical and data mining analysis	Analysis of intensity microarray data to produce final tabular dataset
The GenoMetric Query Language (GMQL)	Free	Query, Download and Integrate Public with Private Genomic Datasets	Integrate Public with Private Genomic Datasets	Web interface	Platform-independent	NO	NO	NO	M	Allows to perform only genometric queries	It combines private dataset with publicly available datasets
PLINK	Free	Analysis of genotype/phenotype data	YES	command line, GUI (5)	Platform-independent	NO	YES	YES	H	Allows to perform only statistical analysis	Data management, statistical analysis, association analysis of whole-genome studies
Haploview	Free	Haplotype analysis	YES	GUI	Platform-independent	NO	YES	NO	M	Allows to perform only statistical analysis	Graphical computation of LD statistics and population haplotype patterns
coreSNP	Free	Parallel analysis of Variants based on Statistical test	NO	GUI	Platform-independent	NO	YES	YES	L	single allelic variants discovery; Unable to dig with Affymetrix raw files	Massive parallel analysis of SNPs dataset
PARES	Free	Parallel association rules extraction from SNP Datasets	NO	GUI	Platform-independent	YES	NO	YES	L	Manage small Dataset	Multiple allelic variants discovery
Genotype Analytics	Free	Web Services for bioinformatics	YES	Web interface	Platform-independent	YES	YES	YES	L		Data mining and statistical Web Services to analyse SNP Datasets
Cloud4SNP	Free	Cloud Serices to analyse SNP Datasets	YES	Web interface	Platform-independent	NO	YES	YES	L	Allows to perform only statistical analysis	Easy to use through Web Browser

Notes: This table summarizes the main features of the listed genomics framework tools. In the table, the column “Availability” provides information about the license of use of each tool. Free* in Availability means that the tool is free to use after registration on the vendor web site. “Main features” column shows the main functionalities of the listed tools. “IA” stands for Integrative Analysis specifying if IA is provided by the tool. “Interface” column provides information about how to access the tool functions. “OS” stands for Operating System, indicating the compatibility between OS and each tool. “DM” is short for Data Mining, meaning if DM analysis is provided by the tool. “Statistical” indicates if Statistical Analysis is provided by the tool. “Prep” stands for Preprocessing and it shows if Preprocessing is provided by the tool. “CS” stands for Computer Skills and it describes the levels of competency in using tools (i.e., L indicates basic computer knowledge, M indicates limited computer knowledge, H indicates expert to manage software).

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
