# Peer review of "DMETTM Genotyping: Tools for Biomarkers Discovery in the Era of Precision Medicine"

_2571-5135, 2020, doi:10.3390/ht9020008_

Round 1

Reviewer 1 Report

This review paper explored all the algorithms developed to preprocess and analyze genotyping data, and therefore it would be a good starting point for beginners to read.  I recommend this review for publication without changes. 

Author Response

We are grateful to the reviewer for manuscript consideration

Reviewer 2 Report

In this review, the authors have done a good job of describing the software/tools used in the DMET genotyping analysis. At first instance, provide a full form of DMET Briefly explain the DMET analysis workflow in the introduction section. You can also provide a flowchart here. Please, provide a detailed comparison for different software/tools mentioned in the manuscript. It will be good to know the advantages and disadvantages of each tool. You can provide this information in table format. Provide the list of the gene names as mentioned in line # 231. It will be good to know which genes are involved in the DMET array. You can provide this list as a supplementary file. Please discuss briefly the biomarker validation in this review. How DMET and GWAS are different? It is necessary to discuss this point in this review. The manuscript has several grammatical mistakes, word spacing, writing genes and protein names, and different fonts sizes. The short gene names should be italic. I recommend a significant revision in the writing of the manuscript.

Author Response

At first instance, provide a full form of DMET . Briefly explain the DMET analysis workflow in the introduction section. You can also provide a flowchart here

Many thanks for this suggestion. In the revised manuscript we explain DMET analysis workflow in the introduction section, lines # 99-112 and included a flowchart (fig.1).

Please, provide a detailed comparison for different software/tools mentioned in the manuscript. It will be good to know the advantages and disadvantages of each tool. You can provide this information in table format.

We thanks to reviewer for the suggestion. In the revised manuscript we add a table (table 1) including comparison for different software/tools mentioned in our manuscript.

Provide the list of the gene names as mentioned in line # 231. It will be good to know which genes are involved in the DMET array. You can provide this list as a supplementary file.

Many thanks to the reviewer for this indication. We included a full list DMET genes in a supplemental material (Table S1)

Please discuss briefly the biomarker validation in this review.

We thanks to reviewer for the comment. We discuss this point in the conclusion.

How DMET and GWAS are different? It is necessary to discuss this point in this review.

We thanks to reviewer for the comment. We discuss the difference between DMET and GWAS in the introduction, lines # 72-79

The manuscript has several grammatical mistakes, word spacing, writing genes and protein names, and different fonts sizes. The short gene names should be italic. I recommend a significant revision in the writing of the manuscript.

The revised version of the manuscript has been implemented following the reviewer suggestions.

Round 2

Reviewer 2 Report

The authors have addressed all the comments.